# Next Generation Mucosal Vaccine Strategy for Respiratory Pathogens

**DOI:** 10.3390/vaccines11101585

**Published:** 2023-10-12

**Authors:** Farokh Dotiwala, Arun K. Upadhyay

**Affiliations:** Ocugen Inc., 11 Great Valley Parkway, Malvern, PA 19355, USA

**Keywords:** mucosal immunity, inhaled vaccines, respiratory infections, COVID-19, influenza, SARS-CoV-2, mRNA, adenovirus, mucosa-associated lymphoid tissue

## Abstract

Inducing humoral and cytotoxic mucosal immunity at the sites of pathogen entry has the potential to prevent the infection from getting established. This is different from systemic vaccination, which protects against the development of systemic symptoms. The field of mucosal vaccination has seen fewer technological advances compared to nucleic acid and subunit vaccine advances for injectable vaccine platforms. The advent of the next-generation adenoviral vectors has given a boost to mucosal vaccine research. Basic research into the mechanisms regulating innate and adaptive mucosal immunity and the discovery of effective and safe mucosal vaccine adjuvants will continue to improve mucosal vaccine design. The results from clinical trials of inhaled COVID-19 vaccines demonstrate their ability to induce the proliferation of cytotoxic T cells and the production of secreted IgA and IgG antibodies locally, unlike intramuscular vaccinations. However, these mucosal vaccines induce systemic immune responses at par with systemic vaccinations. This review summarizes the function of the respiratory mucosa-associated lymphoid tissue and the advantages that the adenoviral vectors provide as inhaled vaccine platforms.

## 1. Introduction

The severe acute respiratory syndrome coronavirus 2 (SARS-CoV-2) pandemic is a reminder that the emergence of novel mucosal pathogens, with no or suboptimal vaccines against them, can cause an unacceptably high strain on global health status and infrastructure. In the past decade, the global scientific community and the pharmaceutical industry have focused on vaccine development for respiratory pathogens [1]. However, most of the recent advances have been in injected vaccines, whereas implementation of mucosal vaccines has lagged. There are only nine FDA-approved mucosal vaccines for human use, eight of which are given orally, and only one is administered intranasally (Flumist—MedImmune/Sanofi Pasteur) [2]. Although these mucosal vaccines are effective, they rely on the use of attenuated or inactivated pathogens and have not yet seen the practical benefits from new vaccine technologies such as mRNA, subunits, or nanoparticles like injectable vaccines have. Convidecia-Air (Ad5-nCoV), an orally inhaled vaccine, and iNCOVACC (BBV154), given by intranasal drops, have both incorporated the benefit of adenoviral vector technology and are currently approved in China and India, respectively [3,4].

The interest in mucosal immunization is revitalized due to its numerous advantages over injectable vaccines [5], which include (1) secretory antibody (IgA) production and (2) the presence of tissue-resident effector and memory T cells at the mucosa [6,7]. (3) Since the mucosa is the entry point for >90% of pathogens—from respiratory viruses to sexually transmitted and enteric diseases—mucosal immunity is the first line of defense against the establishment of initial infection [8]. (4) Since the mucosa also sheds these pathogens, a strong mucosal immunity would prevent transmission in case the infection gets established. Infected individuals vaccinated with injectable vaccines have shown continued transmission of SARS-CoV-2 [9,10]. (5) Induction of mucosal immunity at one site protects other mucosal sites and produces a robust systemic immunity. (6) Oral drop, intranasal drop/inhaled, or orally inhaled mucosal vaccines also provide practical benefits such as needle-free easy administration and better accessibility for underdeveloped regions that suffer from poor logistics and lack of ultra-low cold chain. These advantages of mucosal vaccines have been highlighted by recent preclinical and clinical studies on SARS-CoV-2 mucosal vaccines [11,12,13,14,15].

Lower respiratory tract infections (LRI) are a major cause of global mortality. (WHO 2020) A 2019 global burden of disease study revealed that LRIs were responsible for approximately 489 million infections and 2.5 million deaths per annum across 204 countries [16]. While *Streptococcus pneumoniae*, respiratory syncytial virus (RSV), *Hemophilus influenzae B*, and influenza virus are still among the top causes of LRI particularly in the young (<5 years old) and older people, the COVID-19 pandemic is responsible for a cumulative 766 million cases and 6.9 million deaths worldwide. (WHO COVID Dashboard)

Arexvy from Glaxo Smith Kline is currently the only approved vaccine against RSV for elder patients (≥60 years old), (Arexvy-FDA approval) with Abrysvo from Pfizer receiving the approval for the protection of babies through maternal vaccination in the third trimester (Abrysvo—approval). There are licensed injectable vaccines targeting respiratory pathogens like *S. pneumoniae*, *Mycobacterium tuberculosis* (MTB), *Bordetella pertussis*, and influenza. However, current trends in vaccine development offer promise in enhancing suboptimal protection at the site of infection using mucosal vaccines. Flumist is an intranasal influenza vaccine approved by the FDA in persons 2–49 years of age [17], intranasal administration of Bacillus Calmette–Guérin (BCG) shows promising results in the prevention of MTB infection [18], and BPZE1, an intranasal live attenuated pertussis vaccine, consistently induced a broad pertussis-specific mucosal secretory IgA responses, whereas the conventional injectable Tdap did not [19,20]. The XBB variants of SARS-CoV-2 have considerably reduced the efficacy of the array of the currently implemented COVID-19 vaccines, including bivalent Wuhan1/BA4-5 vaccines [21]. Mucosal delivery could avoid these vaccination issues and enhance convenience and patient compliance [5]. However, so far only China (CanSino—Convidecia Air) and India (Bharat Biotech—iNCOVACC) have approved mucosal vaccines against SARS-CoV-2. The development of combination mucosal vaccines targeting conserved antigens from multiple pathogens like SARS-CoV-2, seasonal influenza viruses, and RSV, although challenging, may be a viable annual/semiannual option for the prevention of future pandemics [22].

## 2. The Mucosa-Associated Lymphoid Tissue

Mucosal immunity arises from the highly organized secondary lymphoid tissues called mucosa-associated lymphoid tissues (MALTs), where antigen-specific immune responses are initiated. In certain mucosal tissues, the immune components lack specific structures and exist as a diffused network of lymphoid and mucous membrane-associated cells along with cytokines, chemokines, and their receptors under the lamina propria, which are induced by infection and produce secretory IgA antibodies (Figure 1). These are called tertiary or ectopic lymphoid tissues. MALTs are further divided according to the mucosal tissues they are situated in, such as the gut-associated lymphoid tissues (GALT), the nasopharynx-associated lymphoid tissue (NALT), the bronchus-associated lymphoid tissue (BALT), the conjunctiva-associated lymphoid tissue (CALT), and the vaginal-associated lymphoid tissue (VALT) (Figure 1). Each MALT is organized like a lymph node with B-cell-rich follicles and T-cell-rich interfollicular areas with antigen-presenting dendritic cells (DC) and antigen-sampling microfold (M) cells in the covering epithelium. High endothelial venules (HEVs) provide entry and egress to lymphocytes that recirculate in the MALT or migrate to other lymphoid tissues, enhancing systemic immunity [23]. In addition, these cells function in synergy with innate immune cells like NK cells, innate lymphoid cells (ILCs), mucosal-associated invariant T (MAIT) cells, and γδ-T cells and anti-microbial molecules like defensins, cathelicidin, lysozyme, mucin, and surfactants. Even within mammals, interspecies differences exist in the structure and regulation of MALT. For instance, unlike rats, both humans and mice do not have an anatomically distinct NALT and BALT. They instead have oropharyngeal and bronchoalveolar lymphoid tissues that are induced by pulmonary infections (Figure 1) [24].

## 3. Immune Surveillance at Mucosal Surfaces

The epithelium over BALT is permeable to the immune crosstalk between the lumen and the lymphoid tissue. The mucosal cells express diverse pattern recognition receptors and allow the use of multiple antigen sampling strategies to detect the infection [25]. These epithelial cells also express antimicrobial effector molecules and help in the transcytosis of secretory IgA antibodies against the infective agent. In tissues such as airway passages and intestines, where tight junctions secure the epithelial intercellular spaces, specialized microfold epithelial cells (M-cells) deliver foreign antigens from the lumen to the MALT (Figure 2) [24]. M-cells have fewer microvilli than the surrounding epithelial cells, instead having short fold-like structures called microfolds. In addition, towards the basal side, the M-cell plasma membrane is deeply invaginated to form a large M-cell pocket where B/T lymphocytes or DCs can reside. The very thin M-cell cytoplasm at the pocket helps in the phagocytosis/transcytosis of antigens from the lumen and allows for easier contact of MALT immune cells with pathogens and better antigen presentation [26]. M-cells have exceptionally low numbers of lysosomes with low enzyme activity [27]. As a result, M-cells do not process the antigens taken up but instead just transfer them intact to DCs, which perform the antigen processing and presentation (Figure 2).

DCs are the true sentinels in MALT, residing in the M-cell pocket, sampling luminal antigens, and then migrating either to the local MALT or the distant draining lymph nodes. The local DCs not only induce immune responses against pathogens but also play an important role in tolerance to antigens from commensal microbiota and food [28]. To help with the tolerogenic functions, the MALT-associated DCs and naïve CD4 T-cells have higher interleukin (IL)-10 secretion than their splenic counterparts [29]. In mice, CD11b+CD8- DCs form the tolerogenic DC component, whereas CD11b-CD8- and CD11b-CD8+ DCs are pro-inflammatory, produce IL-12, and present antigens to and prime IFNγ production from T cells [30]. The CD11b+CD103+ subpopulation of DCs migrates from the lamina propria to the draining lymph nodes in a CCR7-dependent manner [31,32]. Through this migration, the CD103+ DCs present antigens from mucosal sites to CD8+ and CD4+ T cells in the local lymph nodes, resulting in the expression of homing chemokines CCL17, CCL19, CCL21, CCL22, CXCL13, and IL-16 (Figure 2) [31,32]. In mice and humans, this process is responsible for the infection-driven spontaneous BALT formation [32]. In addition, immune complexes with antigen-IgG antibody complexes (IgG-IC) are recognized by the FcRn receptors found abundantly on mucosal DCs. The FcRn-bound IgG-ICs are internalized into DC compartments where the antigen is processed into peptide epitopes compatible with both major histocompatibility complex class I and II (MHCI/II) molecules. This plays a major role in antigen cross-presentation leading to potent mucosal CD4 and CD8 T-cell responses [33].

## 4. Mucosal Humoral Response

A hallmark of mucosal immunity is the production of secretory IgA (sIgA) by activated B-cells in the MALT [34]. The activated B-cell class switching for IgA production is driven by TGF-β, retinoic acid, IL-4, IL-6, and IL-10 [35,36] produced by stromal cells, epithelial cells, DCs, and mucosal lymphocytes [37]. After class switching, IgA-secreting B cells enter circulation via efferent lymphatic vessels and are disseminated to systemic and other mucosal effector tissues as long-lived IgA-producing plasma cells [38]. IgAs are produced as monomers found in serum or as dimeric sIgA secreted at the mucosal lamina propria by the activated MALT B cells. The sIgA dimers are formed by two IgA monomers covalently linked through disulfide bonds and a single 15-kDa joining or J chain and the 18 amino-acid carboxy-terminal extensions of one of the heavy chains of each monomeric IgA [39,40]. In the MALT dimeric sIgA, binds with high affinity to the secretory component (SC) ectodomain of polymeric Ig receptors (pIgR) found on the basolateral surface of epithelial cells. The pIgR-bound IgA is then actively transported across the mucosal epithelium to the lumen where the sIgA-SC complexes are released [39,40]. The sIgA-SC complex is acid and protease-resistant and shows immune interactions that are unique from monomeric IgA or IgG found in serum [36]. Unlike IgGs, IgA antibodies cannot activate the classical complement pathway as they lack C1q binding sites. Therefore, IgAs predominantly serve as neutralizing antibodies, leading to non-inflammatory immune responses that limit damage to the mucosa. The sIgA antibodies protect the mucosa by inhibiting viral or bacterial binding to epithelial cells and subsequent cell entry [41,42]. Pathogens like the influenza virus that require glycan interactions for infection can be blocked by the interaction of the glycans on IgA [43]. Multiple studies report that even during its passage through the mucosal epithelium, sIgA effectively neutralizes viruses like Sendai, influenza, measles virus, rotavirus, HIV, and SARS-CoV-2 [44,45,46,47,48,49]. The sIgA-SC complexes can interact with innate defense machinery like mucins, lysozymes, lactoferrins, and lactoperoxidase [50]. Multiple antigen binding sites of the mucin-anchored sIgA-SC complexes provide high avidity binding to pathogens. This causes pathogen agglutination, a process of forming aggregates that are unable to penetrate through the mucin lining and infect the mucosal epithelium [42,50]. The M cell and epithelium-bound sIgA-SC complex helps in excluding harmful pathogens and toxins from the mucosal surface and favors biofilm formation by commensal microbiome [25]. Secretory IgA is the most abundant and more cross-protective than other immunoglobulin classes [51], can induce antibody-dependent cellular cytotoxicity [52], and has both immune-stimulating [53] and anti-inflammatory properties [54]. Several reports already show the importance of sIgA in immune responses and outcomes for RSV [55], influenza [51], and SARS-CoV-2 [56].

## 5. Mucosal Cellular Immune Response

Central memory T cells (T_CM_) patrol secondary lymphoid organs, whereas T effector memory cells (T_EM_) recirculate between blood and non-lymphoid tissues [57]. Tissue-resident memory T cells (T_RM_) are found locally at multiple MALT sites [58], show distinct cellular signaling and transcription profiles from T_CM_s and T_RM_s, and play important roles in rapid responses to repeat infections [59]. Naïve T cells are activated and undergo clonal expansion after antigen presentation and co-stimulation from antigen-presenting cells (APC) like DCs and macrophages (Figure 2). The APCs also provide cytokine signals that drive the T cell differentiation and homing to different MALT [60]. Effector CD8 T cells are primed in the draining lymph nodes and migrate to the MALT at the site of infection, where they undergo transcriptional reprogramming, which results in elevated expression of T_RM_-specific cell surface proteins CD103 and CD69 [61]. CD103 interacts with E-cadherin and promotes T-cell adhesion with epithelial cells, and CD69 acts as an antagonist of S1PR1 and prevents T_RM_ egress from the respective MALT and its surrounding tissue [62]. T_RM_s surveil the surrounding tissues and rapidly acquire effector CD8 functions on secondary antigenic stimulation [63]. 

T_RM_ cells are known to stably reside in the peripheral tissues without entering the circulation; however, their longevity depends on the tissue of residence. For instance, T_RM_s in the skin are reported to be stable for as long as 300 days post-infection [64], whereas the lung T_RM_ population wanes by 100 days post-influenza infection, leading to a loss in cross-protection against influenza subtypes [65,66]. The lack of lung T_RM_ longevity is not clearly understood and is a concern and a challenge in optimal mucosal vaccine design against respiratory pathogens like flu, RSV, and SARS-CoV-2. Repeated antigen exposure has been shown to increase T_RM_ longevity in mouse models of influenza, likely by upregulating the pro-survival, anti-apoptosis protein Bcl2 [64,65]. In addition to CD69 and CD103, several T_RM_ cell surface markers play a role in their homing and retention in peripheral tissues. T_RM_ in the gut express α4β7 and CCR9, whereas those in the skin express CCR10 [61]. T_RM_ in the lungs and the genital tract express CD49a, with lung-specific T_RM_s also expressing α4β1, CXCR6, and LFA1 [67]. In addition, tissue-released cytokines such as TGF-β, TNF-α, IL-15, and IL-33 are important in maintaining T_RM_ cells.

Due to their constant surveillance at the first exposure sites, T_RM_ cells are an effective and rapid mechanism in restricting pathogen replication (especially intracellular virus replication) at barrier sites like skin, gut, and respiratory system [59,66]. T_RM_ levels in the mouse lungs are a better marker for reinfection protection than memory T cells in the blood [68], hence one school of thought suggests that T_RM_ be used as a benchmark for the testing immunogenicity of mucosal vaccine candidates [69,70]. 

## 6. Inducing Immunity Systemically and at Local Mucosal Sites

Mucosal infections or immunization by mucosal vaccines prime the immune responses that are detectable in both the mucosal and the systemic compartments [71]. Systemic immunization by parenteral routes only induces weaker mucosal immunity [72]. Most mucosal vaccines administered via the oral, intranasal, and inhaled routes induce B and T cells that migrate to systemic secondary lymphoid tissues and even to mucosal tissues different from the site of induction [41,50]. After IgA class switching, activated B cells in the MALT usually elevate CCR10 expression on their surfaces, which favors migration to epithelial cells in the lung, gut, mammary, and salivary glands that express CCL28 [73]. DCs activated at a mucosal immunization site also recirculate to secondary lymphoid tissues to induce T cell priming systemically and at distant mucosal tissues like the vaginal tract [74]. This led to the discovery of prime and pull vaccination to induce immune protection in immunologically restrictive tissues like the genital tract [75]. A robust systemic T cell response can be generated by mucosal immunization (prime), followed by recruitment (pull) of activated T cells either on their own or by topical chemokine application [76]. Therefore, depending on the pathogen’s site of entry, the route of mucosal immunization should be carefully considered.

## 7. Modulating Mucosal Immune Responses by Immunization Routes

Even within mucosal immunization, different routes induce differences in the potency and longevity of the immune response. The oral route usually activates the immune responses in the GI tract, the oral mucosa, NALT, and the mammary glands. Intranasal and inhaled delivery effectively induces immune responses in the salivary glands, the NALT, the BALT, and the lower respiratory and urogenital tract [2,71]. Attenuated or recombinant non-pathogenic viruses or bacteria, administered orally, have been successfully used as antigen vectors to induce robust immune responses against mucosal pathogens and tumors induced by HPV type 16 [77,78]. Similar studies performed using intranasal and inhaled routes of administration report robust immune responses at the lung mucosa with a reduction in transmission [11,12,19,20,79]. A unique feature of the intranasal/inhaled route is the induction of Th17 effector and IL-17-producing T_RM_ cells [80]. Earlier work shows that the Th17 response occurs regardless of the adjuvants used [81]. However, adjuvants can modulate and improve the robustness of this response [82]. Previously associated with detrimental immune outcomes [83], the Th17 responses from nasal immunization [81] and in Peyer’s patches [84] are now considered important for immune responses and pathogen clearance. Due to the stratified keratinized nature of the vaginal mucosa and variations in the local tissue environment from hormonal fluctuations, the intravaginal route is less effective at inducing a local mucosal immune response [85]. As alternatives sublingual and intranasal routes both induced higher numbers of IFNγ-secreting lung CD8 T cells than the intramuscular route [86], but the sublingual route induced less neutralizing antibodies in the genital tract than the intramuscular route [87]. Advancements in vaccine delivery and adjuvant technology continue to improve mucosal immunity [88]. 

## 8. Training Mucosal Innate Immunity

Tolerance to foreign antigens by the mucosal innate immune system is a key reason for infection by SARS-CoV-2, influenza, and RSV, which have short incubation periods. The effector function of innate immune cells can be ‘trained’ by metabolic and epigenetic reprogramming, such that instead of tolerating the vaccine or pathogen antigen, the innate immune system mounts an early response [89,90,91,92]. The trained immunity is reported to last from a few months up to a year [93] and can be induced by microbial or nonmicrobial stimuli. Microbial sources reported to induce trained immunity include the BCG, oral polio, smallpox, and measles vaccine [93,94,95], as well as the malaria parasite *Plasmodium falciparum*, the hepatitis B virus, low-dose lipopolysaccharide (LPS), and the β-glucan component in the cell wall of the fungal pathogen *Candida albicans* [96,97,98]. Trained immunity can also be induced by non-microbial sources like liver X receptor agonists [99], lipoprotein a [100], uric acid [101], aldosterone [102], interferons [103], S100 protein [104], and catecholamines [105]. Multiple labs report improvement of mucosal defense and protection against respiratory infections by successfully induced long-term trained immunity. Examples include (1) BCG vaccine-induced synergistic protection against SARS-CoV-2 [90], yellow fever [106], malaria [107], and typhoid [108]; (2) β-glucan induced protection against *Leishmania* [109], *M. tuberculosis* [110], *Pseudomonas* [111], and tumor metastasis [112]; (3) S100-induced protection from neonatal sepsis [104]; and (4) CpG dinucleotide-induced protection from *E. coli* meningitis [113]. The use of appropriate adjuvants to induce trained immunity in APCs of the mucosa is a promising avenue for future vaccine technology.

## 9. Currently Licensed Mucosal Vaccines

Progress in vaccine technology has shifted injectable vaccines from killed or attenuated whole-cell vaccines towards protein subunit, viral vector, or nucleic acid (DNA or mRNA) vaccines [114,115]. The mucosal vaccine landscape is quite the opposite with the eleven approved mucosal vaccines being either live attenuated, whole-cell inactivated, or adenoviral vector vaccines (Table 1). This is partly because mucosal sites afford high tolerance to whole-cell antigens while rapidly clearing subunit antigens [71]. The mucosal route of vaccination like natural infection produces mucosal and systemic immunity, which is more effective and cross-protective against subsequent infection and transmission prevention than injectable vaccines [116]. In multiple studies, mucosal vaccination provides better protection against mucosal pathogens like influenza, herpes simplex virus, and *Mycobacterium tuberculosis* [117,118,119]. While studies with a head-to-head comparison between mucosal and injectable routes are lacking, reports during the SARS-CoV-2 pandemic show superior humoral and mucosal immune responses with the mucosal route [14,15]. While circulating antibodies induced by injectable vaccines may transduce through at mucosal sites, CD8 T cell homing to mucosal sites and T_RM_ formation are better achieved by mucosal vaccination [61,69,70]. 

## 10. Nucleic Acid and Subunit Mucosal Vaccines

Although nucleic acid technology (DNA/mRNA) has been researched since the 1990s [120], their use in humans was first licensed during the SARS-CoV-2 pandemic. All licensed mucosal vaccines are composed of whole-cell pathogens either alive or dead, and no nucleic acid or subunit mucosal vaccine has been successful so far. This is primarily due to mucosal tolerance to live or whole-cell antigens and because nucleic acid or subunit antigens are susceptible to enzymatic, chemical, or microbiota-imposed degradation (Table 2) [121]. They also suffer from poor penetrance of the mucus layer, rapid degradation of nucleic acids, and poor cellular uptake or transfection efficiency [122]. The latest subunit vaccines have three components: an antigen, an adjuvant, and a targeting system for precise delivery to the right location and at the right time (Figure 3). 

## 11. Synthetic Carriers and Routes of Mucosal Vaccine Delivery

Many effective mucosal delivery systems for nucleic acid and subunit vaccines use nanoscale carriers. These delivery systems include non-viral particles like polymeric or lipid nanoparticles [127], liposomes [128], immune stimulatory complexes (ISCOMS) [129], micelles, microspheres [130], self-assembling peptides [131], proteosomes [132], and lipid–aqueous phase emulsions (Figure 3) [133]. Due to the less dramatic changes in the pH, the nasal route is better suited for nucleic acid and subunit vaccines than the oral route. However, antigen uptake by APCs and subsequent immune response could still be poor due to the profuse mucosal secretions and mucociliary clearance [2]. Advances in nanocarrier biomaterials have initiated the development of protective delivery strategies [124,134]. Some of these polymer biomaterials include polysaccharides (chitosan, β-glucans, cyclodextrin, etc.) [135], polypeptides (poly-L-lysine) [136], polyamines (Polyethyleneimine—PEI) [137], polyesters (Polyhydroxyalkanoate—PHA, poly lactic-co-glycolic acid—PLGA, Poly β-amino ester—PBAE) [138], and polyamidoamine dendrimers with modifications like fluorination [139], glycosylation [140], and PEGylation [141]. These polymers used in different stoichiometry can control various vaccine properties such as surface charge, rigidity, distribution and retention in the lymph nodes, and maturation and uptake by the DCs [134]. 

## 12. Live Attenuated Viruses and Viral Vectors for Mucosal Vaccines

Historically live attenuated vaccines (LAVs) were produced by evolving pathogenic viruses via serial passage through a foreign host species using tissue cultures, chicken eggs, or live animals [142]. Eventually, by selection pressure, the viruses develop mutations that favor their growth in the foreign species and become less virulent to the original host (humans). However, recent vaccine ventures have used codon de-optimization and insertion of key mutations in the SARS-CoV-2 and other viral genomes to produce LAVs for COVID-19 and other viral diseases [143,144]. These modern LAVs contain nearly all the immunogenic antigens preserved in their native conformation. Therefore, live attenuated vaccines are the most immunogenic and have a long history of success at preventing infections (Table 2) [125]. All successful vaccines against systemic viruses such as measles, mumps, rubella, variola (smallpox), and varicella (chickenpox) are live virus-replicating vaccines [145]. These viruses may transmit by aerosolization but replicate systemically and have long incubation periods (≈10–16 days), leading to good exposure to the host mucosal and systemic immune system. Similar successes have been observed with live enteric viral vaccines against polio and rotavirus [146]. However, live attenuated vaccines have a higher risk of reversion to a virulent state than other vaccine types and the danger of infection in patients with HIV or other immunocompromised states such as cancer therapy or transplantation [71,145]. Live attenuated influenza vaccines were tried over the last decade and demonstrated less than adequate protection from infection [147]. However, intranasal vaccination using adenoviral vectored influenza nucleoprotein-induced sustained CD8 T_RM_ cells in mouse lungs for longer than a year [148]. The induced CD8 T_RM_ cell protection in vaccine-treated mice persisted longer than in influenza-infected mice. Viral vectors do not suffer from the same limitations as subunit or nucleic acid vaccines and therefore represent the most promising strategy for mucosal vaccines [71], especially in the respiratory tract. Intranasal delivery of adenoviral vectored SARS-CoV-2 spike antigen-induced protective immune responses in animal models [11,12,13] and human clinical trials [14,149]. While both intranasal and intramuscular routes induced comparable systemic humoral and T cell responses, on SARS-CoV-2 viral challenge, the hamsters vaccinated by intranasal route showed significantly lower viral loads in the nasal tract and lungs than the hamsters that received intramuscular vaccination [11]. Although recombinant human adenoviruses are the predominantly used viral vaccine vectors [150], several other recombinant viruses such as modified vaccinia virus Ankara (MVA—poxvirus) [151], chimpanzee-derived adenoviruses (ChAd) [11,12,13], recombinant rhesus cytomegalovirus (RhCMV) [152], recombinant RSV [153], recombinant lentivirus [154], attenuated influenza or parainfluenza viruses [155,156], and Newcastle disease virus (NDV) [157] have been tested with varying success as viral vectors for mucosal vaccine delivery [5].

## 13. Improving Mucosal Immune Response by Adjuvants

Mucosal vaccine responses can be significantly improved with the use of adjuvants, which can act as delivery vehicles, immunostimulatory molecules, or both (Figure 3 and Table 3). Most adjuvants fall into the Toll-like receptor (TLR) agonist category, enhancing the mucosal immune response and lowering the antigen requirement when co-administered with the antigen [158]. The TLR4 agonist 3-O-desacyl-4′-monophosphoryl lipid A (MPL) is one of the most widely mucosal adjuvants [159] as a combination with QS21 saponin (AS01) or alum (AS04) in several approved intramuscular vaccines like Fendrix (Hepatitis B virus—HBV), Shingrix (Varicella zoster virus—VZV), and Cervarix (Human papillomavirus—HPV) [160,161]. The activation of antigen-presenting and innate immune cells by TLR4 agonists favors Th1 differentiation and strong Th1-associated humoral responses [162]. Multiple studies report the successful generation of robust immune responses by intranasal administration of TLR4 agonists [163,164,165]. Administration of a synthetic TLR4 agonist, Glucopyranosyl Lipid Adjuvant—stable emulsion (GLA-SE)—induced Th1/Th17-biased systemic and mucosal antibody responses when administered intramuscularly [166] or intranasally [167]. Other TLR ligands tested in mucosal vaccines have been (a) TLR3-specific double-stranded RNA analog polyinosine: polycytosine acid (poly I: C) [168,169], (b) TLR5-specific flagellins [170,171], (c) TLR7 agonists like imidazoquinoline derivatives [172], and TLR9 agonists like CpG-oligodeoxynucleotides [173]. The best-studied mucosal adjuvants are bacterial ADP-ribosylating enterotoxins derived from either cholera toxin (CT) or *Escherichia coli* heat labile enterotoxin (LT) [71,174]. Devoid of enterotoxicity, the LT-adjuvanted inactivated intranasal influenza vaccine (Nasalflu-Berna Biotech) was initially approved in Europe but later proved to cause Bell’s palsy in vaccine recipients due to LT accumulation in the olfactory bulb and other nervous tissues [175]. Similar incidences of Bell’s palsy were reported in other vaccine trials [176]. The double mutant LT (R192G/L211A) (dmLT) activates the immune system just as well but without the associated epithelial cell cAMP intoxication or intestinal fluid secretion of LT and has been used in about 25 preclinical studies since 2000 [177]. CTA1-DD adjuvant combines the beneficial immunostimulatory effects of the *V. cholerae* CTA subunit enzyme with the D-domain dimer from *S. aureus* that targets B cells [178]. α-Galactosylceramide activates invariant natural killer T (iNKT) cells and promotes antigen cross-presentation by DC to CD8 T cells [179]. Multiple studies report good efficacy of α-Galactosylceramide adjuvanted live or inactivated intranasal influenza vaccines without antigen redirection to the nervous system and devoid of serious implications to the CNS [180,181]. Chitosan and N-dihydrogalactochitosan nanoparticles are good delivery vehicles due to their high positive charges, which enable strong interactions with antigen molecules and help in crossing biological barriers [182,183]. Being potent activators of macrophages NK cells and lymphocytes, they also serve as adjuvants [184]. Chitosan nanoparticles have been used in the intranasal route to induce high IgA levels in different models [185,186]. 

## 14. Mucosal Vaccine Lessons from Natural Infections

Unlike viruses like measles, mumps, rubella, variola, and varicella, which have long incubation periods (≈10–16 days) and spread systemically to other parts of the body, respiratory viruses like SARS-CoV-2, influenza, and RSV or enteric viruses like norovirus are restricted to the mucosa and have evolved shorter incubation periods (2–5 days) [145]. As the respiratory and gastrointestinal mucosa is exposed to a large variety and amounts of ingested or inhaled foreign antigens (food, dust, pollen, etc.), its innate immune compartments have evolved mechanisms to tolerate transient harmless foreign proteins [89,90,91,92]. As the innate immune system is responsible for the detection and elimination of the pathogen for the first 5–7 days of infection, viruses with short incubation periods like SARS-CoV-2, influenza, norovirus, and RSV undergo unchecked replication at the mucosal sites before the adaptive immune response kicks in [187,188,189,190]. Although the current systemic vaccines effectively reduce the disease severity, the short incubation and rapid viral replication leave a considerable transmission window open before the neutralizing antibody or T-cell response can control the infection. While mRNA vaccines induced IgG titers in the saliva of vaccinated people [191], their neutralizing efficacy and ability to prevent transmission are not known. On the other hand, adenoviral vector vaccines administered intranasally are reported to induce systemic and mucosal IgA and prevent transmission in multiple preclinical models [11,12,13]. Similar immune responses were observed in human clinical trials [14,15]. Hybrid models of vaccination showed successful immune responses and protection in animals primed with the systemic mRNA (or other) vaccines followed by an intranasal booster with the SARS-CoV-2 spike protein [192] or adenoviral vaccines expressing SARS-CoV-2 spike [193,194]. The creation of next-generation vaccines demands a better understanding of different aspects of mucosal immunity, such as sensing at the mucosal epithelium [195,196], tolerance mechanisms in the innate immune system [89,92], the importance of IgA class switching, IgA secretion [197,198] and T_RM_ surveillance, and the role of microbiota in vaccine response [199,200].

## 15. Relevance of Mucosal Vaccination against Respiratory Pathogens

Respiratory viruses such as influenza, RSV, and SARS-CoV-2 first infect the upper respiratory tract, then spread to the lower respiratory tract and the alveoli causing viral or secondary bacterial pneumonia which is the principal cause of death [201,202,203]. Secretory mucosal immunity is more effective than systemic immunity at neutralizing incoming viruses while the T_RM_ cell surveillance rapidly responds to virus-infected mucosal cells [58]. Secretory IgA is effective at preventing viral spread from the upper respiratory tract [204] while neutralizing IgG titers are required to control the infection in the lungs [71]. The sIgA levels in the upper respiratory tract are the best correlates of protection against RSV transmission, with similar results reported from SARS-CoV-2 studies [205,206]. A key roadblock in successful mucosal vaccination is the innate immune tolerance afforded to all antigens in the mucosa. Recent scientific breakthroughs have opened avenues to develop trained immunity-based vaccines that use ligands of pathogen recognition receptors (PRR) found on APCs. Many PRR ligands such as LPS, flagellin, β-glucans, chitin, FimH, muramyl dipeptide, and CpG oligodeoxynucleotides show experimental evidence of cross-protection against mucosal pathogens [207]. 

## 16. Advantages of Adenoviral Vectors in Mucosal Vaccination

Vaccination is integral to infection prevention due to the high mutation rate in SARS-CoV-2. Over the last 2–3 years, tested COVID-19 vaccines have used several modified viral vectors, such as adenoviruses, vaccinia, measles, herpes viruses, rhabdoviruses, influenza, and lentiviruses. Viral vectors have high transduction efficiency in a wide variety of cell types and produce high levels of the target antigens. However, large-scale viral vector use is limited by several factors, such as safety concerns, reproducibility, immunogenicity, potential carcinogenicity, and inflammation against the viral vector [71,145]. Lentivirus vectors, once infected, cannot be eradicated from the body [208], whereas measles and herpes vector vaccines are cytotoxic [209]. The adenoviral vector vaccine has been used effectively in preclinical [11,12,13] and clinical [14,15,149,210] settings as a vaccine against SARS-CoV-2. The adenoviruses are thermostable and induce moderate innate immunity without the presence of adjuvants (Figure 4 and Table 2). As replicating adenoviruses naturally cause respiratory infections, they are already optimal at penetrating the mucus layers in the airways and transducing a variety of cells in the mucosa. The Ad5-nCoV adenoviral vector vaccine was the first SARS-CoV-2 vaccine to be administered as an inhaled aerosol. This route was found to induce a strong humoral and cellular response at 1/5th the intramuscular dose and protected both upper and lower respiratory pathways from SARS-CoV-2 infection [15]. The aerosolized, inhaled Ad5-nCoV blocked SARS-CoV-2 replication in the respiratory tract, thereby preventing person-to-person transmission, an advantage not seen from the intramuscular injection. In addition to these advantages, adenoviral vectors can accommodate large or multiple genes thereby allowing multicistronic expression of different antigens on the same vector [211]. This contributes to the low cost of manufacturing and relatively easy production scale-up to meet vaccine demands. An extremely rare (0.3–1.5/100,000 vaccinations) adverse event called vaccine-induced immune thrombotic thrombocytopenia (VITT) occurs shortly after initial IM vaccination with adenoviral vector COVID-19 vaccines [212,213,214]. VITT is characterized by elevated anti-platelet factor 4 (PF4) antibodies and D-dimer levels, with signs of thrombosis, particularly in the cerebral venous sinus [215,216]. Studies demonstrate that VITT occurs due to intravenous administration of adenovirus or its leakage into the circulation following microvascular injury [217,218,219,220,221]. Although under investigation, it is unlikely that aerosolized inhaled adenoviral vaccine would leak into circulation and cause VITT. Published Ad5-nCoV phase 3 and 4 studies do not report any cases of VITT [149,210]. Vaccines delivered by inhalation show good drug absorption, fast immune response, and high bioavailability [222]. They can be administered with little healthcare infrastructure and eliminate the fear of needles as a reason for poor patient compliance [223]. Higher efficacy than the intramuscular route and intranasal sprays [224] mean that mass vaccination can be achieved at a lower cost.

## 17. Future Perspectives

Mucosal vaccines show promise in both disease treatment and transmission interruption. The pre-COVID mucosal vaccine development had stalled and was limited to traditional forms of antigen delivery such as inactivated and attenuated pathogen vaccines. The different mucous membrane barriers may be responsible for this limitation. The COVID-19 pandemic has reignited interest in the development of mucosal vaccines. The primary focus of respiratory vaccines is to induce a robust and durable immune response by (a) delivering the antigen through the mucus layers, (b) targeting the antigen delivery to antigen-presenting cells (APCs) such as M cells, DCs, and macrophages, (c) inducing a robust antigen expression and presentation by the APCs, (d) driving a robust humoral (secretory IgA) and cytotoxic (CD8+) immune response with durable local memory (T_RM_ cells), and (e) reducing adverse events seen in vaccination by systemic route. In this review, we discuss scientific progress achieved in various aspects of respiratory mucosal vaccines. We believe that mucosal vaccine technology would be key in not only controlling the transmission of future pandemics like SARS-CoV-2 but also in controlling seasonal influenza and RSV, which disproportionately impact the young and elderly populations.

## Figures and Tables

**Figure 1 vaccines-11-01585-f001:**
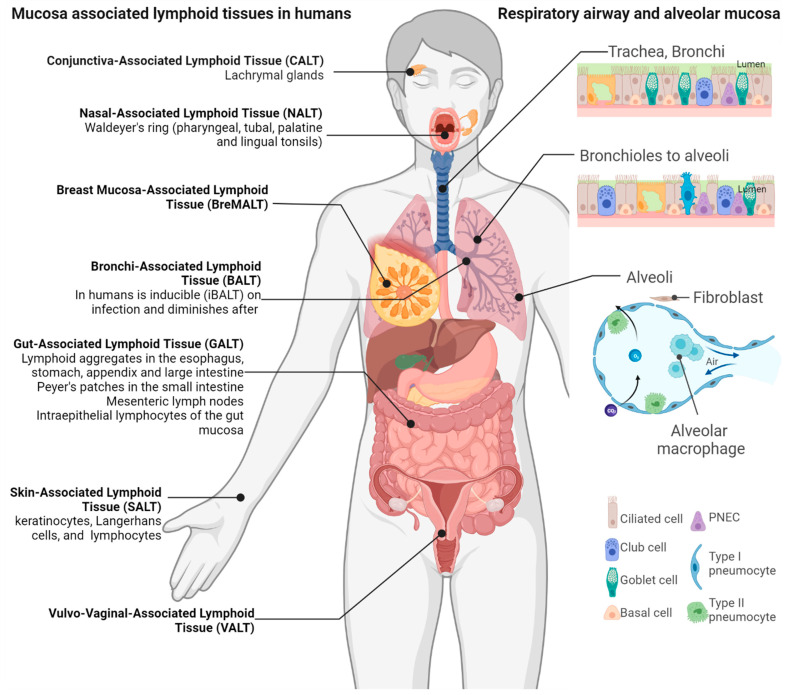
**Mucosa-associated lymphoid tissues**. The immune system associated with various mucosal surfaces constantly surveils these points of entry for pathogens. Humans do not have a defined bronchi-associated lymphoid tissue (BALT), but rather the BALT is induced upon infection. The upper and lower respiratory tract mucosa includes undifferentiated basal cells, ciliated epithelial, pulmonary neuroendocrine (PNEC), and secretory goblet cells. The mucus secreted by the goblet cells provides a barrier to debris, allergens, and potential pathogens but also to mucosal vaccines. Created with BioRender.com.

**Figure 2 vaccines-11-01585-f002:**
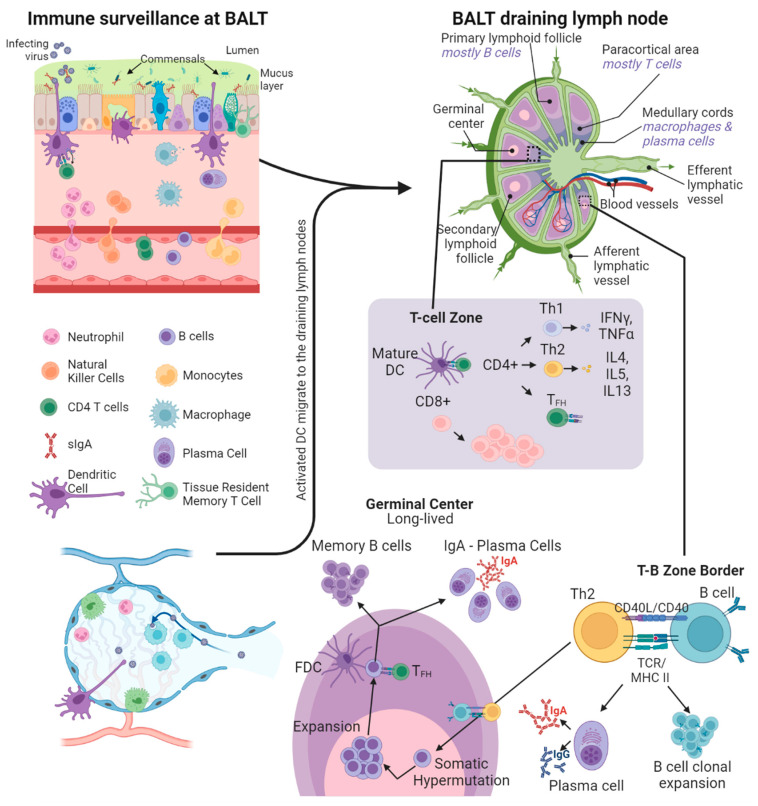
**Immune surveillance at the respiratory mucosa**. Within the mucus layer are innate immune factors such as antimicrobial peptides, proteases, complement system factors, and secretory IgA and IgM antibodies. The dendritic cells (DCs) in the airways become activated on antigen capture and traffic to draining lymph nodes via afferent lymphatics. In T cell zones, the DCs present the antigens on MHC class II receptors to CD4+ T cells and on MHC class I receptors to CD8+ T cells, along with CD80/CD86 co-stimulation. This antigen presentation by DCs promotes the maturation and expansion of naive CD4+ and CD8+ T cells. CD4+ T cells with Th1 polarization assist in the maturation of cell-based cytotoxic immune responses, whereas Th2 polarized CD4+ T follicular helper (T_FH_) cells migrate to the T-B zone border and assist in the maturation of B cells by TCR-MHC II engagement and CD40/CD40L co-stimulation. The T-B cell pairing causes B cell migration to the germinal centers and their clonal expansion. Within the germinal centers, activated B cells (assisted by follicular dendritic cells (FDC) and T_FH_) undergo somatic hypermutation followed by further expansion. Through this iterative cycle B cells with high affinity to the target antigen are selected, followed by class-switching to either plasma cells or memory B cells, which traffic back to the site of infection in the respiratory mucosa. Created with BioRender.com.

**Figure 3 vaccines-11-01585-f003:**
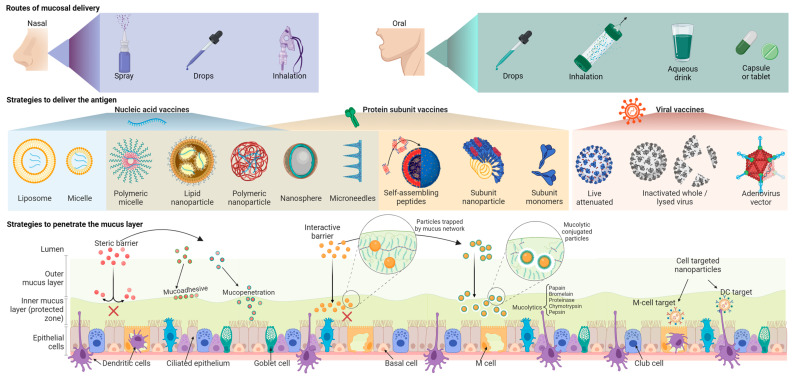
**Strategies to improve the delivery, penetration, uptake, and immunogenicity of respiratory mucosal vaccines**. Vaccines targeting the respiratory mucosa can be delivered nasally (drops, spray, or inhalation) or orally (drops, oral inhalation, aqueous drink, or capsule/tablet). The target antigens can be delivered as nucleic acid or protein subunit vaccines packaged in different lipid/non-lipid nanoparticles or by traditional live-attenuated or inactivated whole virus vaccines. Post-COVID, the adenoviral vector system has gained interest as an intranasal/inhaled vaccine delivery platform. Facilitating antigen delivery by nanoparticles involves mucoadhesive, mucopenetrating, or mucolytic strategies. Created with BioRender.com.

**Figure 4 vaccines-11-01585-f004:**
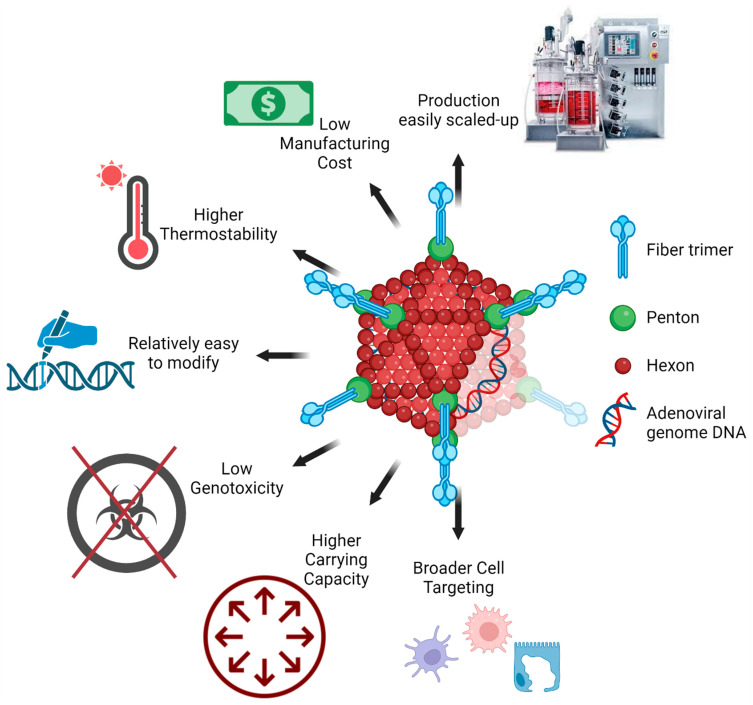
**Advantages of adenoviral vectors for mucosal vaccines**. Naturally existing adenoviruses cause respiratory tract infections and therefore can overcome the mucosal barrier. Adenoviral vectors are capable of transducing multiple types of cells and have a high carrying capacity for large or multi-cistronic antigens. Compared to other viral vectors, adenoviruses are easy to modify, have greater thermostability, and are easy to scale up for mass production. Created with BioRender.com.

**Table 1 vaccines-11-01585-t001:** Currently approved commercial mucosal vaccines.

Vaccine	Composition	Pathogen Targeted	Mucosal Route	Approval Year	Approving Authority
Bivalent Oral Polio Vaccine (bOPV)	Serotypes 1 and 3 attenuated by serial culture	Poliovirus	Oral drops	1961	FDA
Dukoral	Heat and formaldehyde inactivated O1 (Ogawa and Inaba) with recombinant B subunit of cholera toxin	*Vibrio cholerae*	Oral drink	2003	Canada
Shanchol	Bivalent heat and formaldehyde inactivated O1 (Ogawa and Inaba) and O139 serotypes		Oral drops	2013	WHO
Vaxchora	Live O1 Inaba serogroup attenuated by deleting the catalytic domain of the ctxA gene		Oral drink	2015	FDA
Vivotif	Live Ty21a strain attenuated by mutations in LPS and Vi polysaccharide synthesis genes	*Salmonella typhi*	Oral capsule	2013	FDA
Rotateq	Live pentavalent reassortant rotaviruses, containing G1, G2, G3, G4, and P1A strains	Rotavirus	Oral drops	2006	FDA
Rotarix	Live monovalent passage attenuated G1 rotavirus strain with P1A expression		Oral Drops	2008	FDA
Adenovirus vaccine (types 4 and 7)	Live Adenovirus type 4 and type 7 strains	Acute Ad4 and Ad7 respiratory disease	Oral-2 tablets	2011	FDA
FluMist	Quadrivalent live attenuated (cold-adapted) flu A and B strains	Seasonal Influenza	Nasal-Spray	2003	FDA
iNCOVACC	ChAd36 adenoviral vector expressing the SARS-CoV-2 Spike protein (Wuhan)	SARS-CoV-2	Intranasal drops	2022	Central Drugs Standard Control Organization—India
Convidecia Air	Ad5 adenoviral vector expressing the SARS-CoV-2 Spike protein (Wuhan)		Inhaled aerosol	2022	National Medical Products Administration of China

**Table 2 vaccines-11-01585-t002:** Advantages and Disadvantages of different vaccine technologies in mucosal vaccination.

Antigen Delivery	Advantage	Disadvantage	References
**mRNA**	Synthetic, non-infectious, and free from cellular or egg proteins.	Sensitive to pH and degradation by enzymes.	[71,123,124]
	Short development and manufacturing time.	Inability to penetrate mucus barriers.	
	Produces high systemic antibody titers.	Adjuvants are required to break mucosal immune tolerance.	
	Transient expression.	Poor mucosal immune response.	
	Cannot modify host genome.	Ultra-low cold chain required for storage.	
**Protein Subunit**	Can be lyophilized for good environmental stability.	High antigen requirement.	[71,123]
	Can be used regardless of age or immunocompromised status	Sensitive to pH and degradation by enzymes.	
	Cannot modify host genome.	Inability to penetrate mucus barriers.	
		Adjuvants are required to break mucosal immune tolerance.	
		Poor mucosal immune response.	
		Complex manufacturing requirements (conjugation chemistry).	
		Difficult to isolate the most relevant antigens.	
**Live Viral**	Well-established technology. Better stability than mRNA vaccines.	Complex manufacturing and safety requirements.	[71,123,125]
	Naturally capable of penetrating mucus barriers, tolerating high/low pH, and infecting target cells.	Cannot be given to immunocompromised patients.	
	Induces strong mucosal and systemic immune responses.	Small chance of reverting to a pathogenic form and causing disease.	
	May not need adjuvants	Takes time to develop.	
	Simple to manufacture.		
**Viral Vector**	Induces strong mucosal and systemic immune responses.	Concerns for host genome modification/integration.	[71,123,126]
	Naturally capable of penetrating mucus barriers, tolerating high/low pH, and infecting target cells.	Complex manufacturing and safety requirements.	
	May not need adjuvants. Better stability than mRNA vaccines.	Response reduced due to pre-existing immunity against the vector.	
	Simple to manufacture.	Takes time to develop.	
	Cannot cause diseases like live attenuated viruses.		

**Table 3 vaccines-11-01585-t003:** Improving the mucosal immune response involves adjuvant strategies that are different from those used by systemic vaccines.

Class	Molecule/Mechanism	Immune Cell Target	Patents
**Bacterial Toxins**	Double-mutant Labile Toxin	Dendritic cell, Macrophages, M cells	US6033673A
	Cholera Toxin	Dendritic cell, CD4+ T cells	WO2001062283A2
	Cholera Toxin A1-dimer D-domain (*S. aureus*)	Dendritic cell, Macrophages, CD4+ T cells	US8834898B2
**α-Galactosylceramide**	CD1 binding	Dendritic cell, CD8+ T cells	WO2007007946A1
**TLR ligands**	MPL—TLR4	Dendritic cell, Macrophages	US20170182152A1
	CpG—TLR9	B cells, Plasma cells	US6589940B1
	Flagellin—TLR5	Dendritic cell, Macrophages	US7404963B2
**Cytokines**	IL-1, IL-12, IL-18, GM-CSF, RANTES	CD8+ T cells, B cells—IgA, Monocytes, Natural Killer cells, CD4+ and CD8+ T cells	US6168923B1
			US5800810A
			EP1075275A1
			US5679356A
**Chitosan**	Mucoadhesive, improves antigen uptake	Dendritic cell, Macrophages, Natural Killer cells	CN107648603B

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
