# Peer review of "Next Generation Mucosal Vaccine Strategy for Respiratory Pathogens"

_vaccines, 2023, doi:10.3390/vaccines11101585_

Round 1

Reviewer 1 Report

In this manuscript, Dotiwala & Upadhyay give a very clear and through introduction of the state-of-the-art in mucosal vacunación. Sections 2-9 are very well written and helps the reader to understand the key concepts needed for mucosal vaccination to work. These sections are short enough but the information was properly chosen. Sections 10-11 don’t contribute much to the knowledge and I would suggest to remove it as there is not details about vaccines using those platforms. In general, the article is a good review; however, is too long and some sections are very repetitive.

Minor comments
1. Genus and species must be in italics (e.g. lines 56 and 57 Streptococcus pneumonie Hemophilus influenzae type B, ). This mistake repeats over the entire manuscript.
2. Please double check if the recommendation for Abrysvo was moved to approved. I should have gone through approval last week.
3. I would suggest to add a reference at the end of line 127.
4. Line 186. Do you mean adhesion or binding/cell entry?
5. Line 294. It would be better to remove “lab preport” for “reports”.
6. Figure 3. I suggest to move the adjuvant table down below within the figure to increase its size. It is really hard to read.
7. Section 13. It will be useful to list which adjuvants are patented, because most of these adjuvants are not available for most preclinical trials that are not sponsored by the companies that own these technologies.
8. Section 14. Please define  what do you mean by long incubation Times.

Major comments
1. Line 369. The idea is there but that is not really how a live attenuated vaccine is made. The genome is not pathogenic, the virus is pathogenic. Also the mechanism describe there is not the correct one.
2. Line 372. The viruses listed there are not respiratory viruses.

Author Response

We thank the reviewer for his/her time and effort. This critical evaluation of our manuscript has significantly elevated the quality of our manuscript.  Our changes in the new manuscript version are highlighted.

Query: Sections 10-11 don’t contribute much to the knowledge and I would suggest to remove it as there is not details about vaccines using those platforms.

Response: We have merged sections 10-11 into a new section 10 and have reduced the size to one-third of the original sections. (lines 316-326)

Query: In general, the article is a good review; however, is too long and some sections are very repetitive.

Response: We have gone through the entire manuscript and reduced the repetitions and we have reduced redundant references by a third.

Minor comments
1. Genus and species must be in italics (e.g. lines 56 and 57 Streptococcus pneumonie Hemophilus influenzae type B, ). This mistake repeats over the entire manuscript.

Response: Changes made and highlighted on lines 56, 64, 294, 296, 309, 416, and table 1.

2. Please double check if the recommendation for Abrysvo was moved to approved. I should have gone through approval last week.

Response: Lines 60-63 were changed to reflect this, and the hyperlinks were edited.

3. I would suggest to add a reference at the end of line 127.

Response: Added and highlighted (it is line 125 now)

4. Line 186. Do you mean adhesion or binding/cell entry?

Response: We have changed that line to read “binding to epithelial cells and subsequent cell entry” (line 184)

5. Line 294. It would be better to remove “lab preport” for “reports”.

Response: Change highlighted (line 290).

6. Figure 3. I suggest to move the adjuvant table down below within the figure to increase its size. It is really hard to read.

Response: We have separated the adjuvant table from Figure 3 and moved it to the new table 3 (line 393).

7. Section 13. It will be useful to list which adjuvants are patented, because most of these adjuvants are not available for most preclinical trials that are not sponsored by the companies that own these technologies.

Response: In the newly created Table 3 (line 393), we have included patent information in the form of hyperlinks to the respective patents.

8. Section 14. Please define  what do you mean by long incubation Times.

Response: Added and highlighted (line 437).

Major comments
1. Line 369. The idea is there but that is not really how a live attenuated vaccine is made. The genome is not pathogenic, the virus is pathogenic. Also the mechanism describe there is not the correct one.

Response: We have modified the section (lines 357-364) to reflect both the historical and the new reverse genetics methods of viral attenuation with new references. We hope this brings more clarity to the section.

2. Line 372. The viruses listed there are not respiratory viruses.

Response: We have changed the section from lines 366 to 371 and highlighted them. Instead of reading as “These respiratory viruses”, it now reads as “These viruses may transmit by aerosolization but replicate systemically…”.

We thank both reviewers for their time and effort and hope that these changes have adequately addressed your concerns in recommending our manuscript for publication.

Best Regards

Farokh J. Dotiwala

Reviewer 2 Report

The article is well written and informative. However, more information required in relation to nanoparticle delivery systems, different types, adv/disadv for mucosal delivery, uses in vaccine candidate (protein, peptide, mRNA etc) delivery, examples

Author Response

We thank the reviewer for his/her time and effort. The sections added or modified because of this evaluation has significantly improved our manuscript's quality. Our changes in the new manuscript version are highlighted.

Query: However, more information required in relation to nanoparticle delivery systems, different types, adv/disadv for mucosal delivery, uses in vaccine candidate (protein, peptide, mRNA etc) delivery, examples

Response: We have added a new Section 11 (lines 332-349) that discusses synthetic carriers for nucleic acid and subunit mucosal vaccines. We have also included a new Table 2 which compares the vaccine types using their advantages and disadvantages (line 330).

We thank both reviewers for their time and effort and hope that these changes have adequately addressed your concerns in recommending our manuscript for publication.

Best Regards

Farokh J. Dotiwala

Round 2

Reviewer 2 Report

Authors have updated the manuscript based on reviewer comments